



# Automated Avalanche Terrain Exposure Scale (ATES) mapping - Local validation and optimization in Western Canada

John Sykes[1,2], Håvard Toft[3,4], Pascal Haegeli[5], Grant Statham[5,6]

[1]Geography Department, Simon Fraser University, Burnaby, British Columbia, Canada
[2]Chugach National Forest Avalanche Center, Girdwood, AK, USA
[3]Norwegian Water Resources and Energy Directorate, Oslo, Norway
[4]Center for Avalanche Research and Education, UiT the Arctic University of Norway, Tromsø, Norway
[5]School for Resource and Environmental Management, Simon Fraser University, Burnaby, British Columbia, Canada
[6]Parks Canada, Banff, Alberta, Canada

Correspondence to: John Sykes (john_sykes@sfu.ca)

**Abstract.** The Avalanche Terrain Exposure Scale (ATES) is a system for classifying mountainous terrain based on the degree of exposure to avalanche hazard. The intent of ATES is to improve backcountry recreationist's ability to make informed risk management decisions by simplifying their terrain analysis. Access to ATES has been largely limited to manually generated maps in high use areas due to the cost and time to generate ATES maps. Automated ATES (AutoATES) is a chain of geospatial models which provides a path towards developing ATES maps on large spatial scales for relatively minimal cost compared to manual maps. This research validates and localizes AutoATES using two ATES benchmark maps which are based on independent ATES maps from three field experts. We compare the performance of AutoATES in two study areas with unique snow climate and terrain characteristics; Connaught Creek in Glacier National Park, British Columbia, Canada and Bow Summit in Banff National Park, Alberta, Canada. Our results show that AutoATES aligns with the ATES benchmark maps in 74.5 % of the Connaught Creek study area and 84.4 % of the Bow Summit study area. This is comparable to independently developed manual ATES maps which on average align with the ATES benchmark maps in 76.1 % of Connaught Creek and 84.8 % of Bow Summit. We also compare a variety of DEM types (LiDAR, stereo photogrammetry, Canadian National Topographic Database) and resolutions (5 m - 26 m) in Connaught Creek to investigate how input data type affects AutoATES performance. Overall, we find that DEM resolution and type are not strong indicators of accuracy for AutoATES, with map accuracy of 74.5 % ± 1 % for all DEMs. This research demonstrates the efficacy of AutoATES compared to expert manual ATES mapping methods and provides a platform for large-scale development of ATES maps to assist backcountry recreationists in making more informed avalanche risk management decisions.

## 1 Introduction

Snow avalanches are a complex and dynamic natural hazard in cold mountainous regions worldwide. In North America and
Europe, on average approximately 140 people are killed by avalanches each year (Avalanche Canada, 2022; Colorado Avalanche Information Center, 2020; Techel et al., 2016), with most of those fatalities being individuals recreating in





mountainous terrain such as backcountry skiers or snowboarders, snowmobilers, mountaineers, and hikers. In most avalanche accidents and fatalities, the victim(s) is a member of the group that triggered the avalanche (Schweizer and Lütschg, 2001; Jamieson et al., 2010).

Avalanche forecast centers exist in many economically developed countries to provide avalanche risk management information to the public in the interest of decreasing the rate of avalanche accidents. Forecast centers typically provide daily reports describing the avalanche danger level as well as detailed descriptions of specific types of avalanche problems that could be triggered (Statham et al., 2018, 2010). However, forecast products generally do not provide information on specific locations that are safe for the current conditions. That decision is left to the end user, who can control their risk exposure by choosing

terrain that is more or less likely to produce an avalanche under the current conditions.

In 2006, the Avalanche Terrain Exposure Scale (ATES) was developed by Parks Canada to provide a systematic approach to classifying avalanche terrain and assist backcountry recreationists in making appropriate and informed risk management decisions. ATES is broken into two models, the communication model and the technical model. The communication model provides accessible and simple descriptions of the ATES classes for a general audience. Each ATES class is defined based on

the slope angle, common terrain characteristics, and level of avalanche exposure. The technical model is oriented towards avalanche professionals or avalanche education audiences who are interested in detailed analysis of avalanche terrain based on an extensive list of terrain characteristics. The original ATES uses 11 terrain characteristics to classify avalanche terrain into a three level scale (Statham et al., 2006). An updated version, ATESv2, uses a simplified list of 8 terrain characteristics to classify avalanche terrain into an expanded 5 level scale (Table 1) (Statham and Campbell, 2023).

In practice, the ATES system has been used to classify linear routes based on the highest level of avalanche terrain exposure along a given route and as zonal polygons where an entire drainage or mountain region is broken into polygons with similar terrain characteristics and assigned an ATES rating (Statham et al., 2006; Campbell and Marshall, 2010). Each of these use cases is relevant for different applications depending on the local terrain and the usage patterns of backcountry recreationists. For example, snowmobile trails, cross country ski trails, or ice climbing routes that travel through avalanche terrain but remain

on a known and established route are well suited to a linear ATES rating. Whereas activities like backcountry skiing or mountain snowmobile riding often involve travel on less defined routes within a drainage or valley which is better suited to zonal ATES mapping. Examples of both linear and zonal ATES ratings can be found across Canada (Avalanche Canada, 2023; Parks Canada, 2023), in guidebooks for selected locations in Canada and the United States, as well as in selected locations in Europe like the Aran Valley in the Pyrenees, Jura Mountains in Switzerland, and in Norway (Gavaldà et al., 2013; Pielmeier

et al., 2014; Larsen et al., 2020).






**Table 1: ATESv2 Technical Model from Statham and Campbell, 2023.**

| | 0 – Non-Avalanche* | 1 – Simple | 2 – Challenging | 3 – Complex | 4 – Extreme |
|---|---|---|---|---|---|
| **Exposure** | No known exposure to avalanche paths | Minimal exposure crossing low-frequency runout zones or short slopes only | Intermittent exposure managing a single path or paths with separation | *Frequent exposure to starting zones, tracks or multiple overlapping paths* | Sustained exposure within or immediately below starting zones |
| **Slope angle** *and* **Forest density** | Very low-angle (< 10°) open terrain or steeper areas of dense forest | Low-angle (< 20°) terrain or steeper slopes in dense forest with openings for runout zones or short slopes | Moderate-angle (< 30°) open or gladed terrain with some open slopes or glades > 35° | Moderate to high-angle (< 35°) terrain with a large proportion of open slopes > 35° and some isolated glades or tree bands | High-angle, open terrain averaging > 35° with a large proportion of slopes > 45° and few or no trees |
| **Slope shape** | Straightforward, flat or undulating terrain | Straightforward undulating terrain | Mostly planar with isolated convex or unsupported slopes | Convoluted open slopes with intricate and varied terrain shapes | Intricate, often cliffy terrain with couloirs, spines and/or overhung by cornices |
| **Terrain traps** | No avalanche-related terrain traps | Occasional creek beds, tree wells or drop-offs | Single slopes above gullies or risk of impact into trees or rocks | Multiple slopes above gullies and/or risk of impact into trees, rocks or crevasses | Steep faces with cliffs, cornices, crevasses and/or risk of impact into trees or rocks |
| **Frequency-magnitude** (events: years) | *Never (> 1:300)* | *< 1:100 - 1:30 ≥ Size 2* | 1:1 < Size 2 *1:30 - 1:3 ≥ Size 2* | 1:1 < Size 3 *1:1 ≥ Size 3* | 10:1 ≤ Size 2 > 1:1 > Size 2 |
| **Starting zone size and density** | No known starting zones | Runout zones only except for isolated, small starting zones with < Size 2 potential | Isolated starting zones with ≤ Size 3 potential or several start zones with ≤ Size 2 potential | Multiple starting zones capable of producing avalanches of all sizes | Many very large starting zones capable of producing avalanches of all sizes |
| **Runout zone characteristics** | No known runout zones | Clear boundaries, gentle transitions, smooth runouts, no connection to starting zones above | Abrupt transitions, confined runouts, long connection to starting zones above | Multiple converging paths, confined runouts, connected to starting zones above | Steep fans, confined gullies, cliffs, crevasses, starting zones directly overhead |
| **Route options** | Designated trails or low-angle areas with many options | Numerous, terrain allows multiple choices; route often obvious | *A selection of choices of varying exposure; options exist to avoid avalanche paths* | Limited options to reduce exposure; avoidance not possible | No options to reduce exposure |

*\* The use of Class 0 is optional due to the reliability needed to make this assessment; otherwise, Class 1 can include Class 0 terrain.*

One challenge of creating ATES maps is the amount of time and cost required for human experts to determine ATES ratings. Efforts to develop a desktop based ATES mapping workflow have helped to decrease the time and cost by using geospatial tools to create preliminary maps which are then refined by local experts (Campbell and Gould, 2013). However, this process largely relies on manual interpretation of geospatial data to determine ATES ratings, so costs are still too high to create ATES ratings for large swaths of mountainous terrain. Hence, the development of ATES maps has so far been limited to high traffic areas where the costs are justified relative to the large number of backcountry users. However, a fully automated approach is required to sufficiently reduce the cost and time necessary to apply ATES on a larger scale and make terrain ratings feasible for more areas.

Automated ATES (AutoATES v1.0) was originally developed by Larsen et al. (2020) using Geographic Information System (GIS) modeling to classify terrain into ATES ratings based on terrain characteristics including slope angle, location of avalanche release areas, and avalanche runout distance. This research showed the ability of AutoATES v1.0 to capture the critical terrain features that are incorporated in manual ATES ratings and highlighted some differences in the scale and resolution of human versus automated maps. The ability to process AutoATES v1.0 for the entire mountainous area of Norway





(365,246 km$^2$) demonstrated vastly improved cost effectiveness for large scale mapping compared to manual methods which warrants continued development and validation of the model.

To utilize its full potential, AutoATES needs to be able to generate ATES ratings for any mountainous region worldwide using locally available input datasets. Some key challenges are evaluating the impact of different snow climates (Mock and
Birkeland, 2000) and determining how different types of DEM and forest input data affect the output of AutoATES (Bühler et al., 2011). Regional differences in snow climate impact typical avalanche types (Haegeli and McClung, 2007) and how snow cover alters the shape of the landscape (Veitinger et al., 2014), which have implications for identifying avalanche release areas and simulating avalanche runout. High quality validation data are critical for testing AutoATES in different snow climates and with different input datasets. Despite fundamental differences in how automated versus manual methods generate ATES
ratings, manual maps created by local experts are the best available dataset for AutoATES validation. Further, the potential to localize AutoATES maps based on expert derived manual maps by fine tuning input parameters for specific regions can improve their performance beyond what is possible using theoretically derived input parameters.

This manuscript presents an approach for localizing and validating AutoATES based on comparison with manual ATES maps and tests the impact of different types of DEM data on the performance of AutoATES, which is part of a broader effort to
update standards for ATES mapping. While detailed information on the development of the ATESv2 classification system can be found in Statham and Campbell (2023), we use a grid search method (Bühler et al., 2018) to optimize the AutoATES input parameters for our study areas using manual ATES maps to assess accuracy. Our study areas are located in mountain ranges with distinct snow and avalanche climates, enabling comparison of optimal input parameters for different regions.

## 2 Methods

### 2.1 Study areas and input data

To validate and optimize the AutoATES model, we selected two test sites in mountain ranges with distinct avalanche and topographic characteristics in western Canada (Figures 2 & 3). Utilizing multiple test sites provides the opportunity to compare the optimal input parameters for different areas. The availability of input data varies by region which is a critical consideration when applying the AutoATES model.

### 2.1.1 Connaught Creek, Glacier National Park, BC

Connaught Creek (Figure 1) is in the Selkirk Mountains, British Columbia, within the boundaries of Glacier National Park. Elevations in the study area range from approximately 1300 to 2700 m with generally steep rocky terrain at higher elevations and densely forested terrain at lower elevations. Connaught Creek is exposed to overhead avalanche hazard from both sides of the valley which makes assessing avalanche terrain exposure challenging.





The area is characterized by a transitional snow climate with strong maritime influence (Hägeli and McClung, 2003). The seasonal avalanche character depends on the strength of the maritime influence. The most important types of persistent weak layers are facet-crust combinations, which typically form after early season rain-on-snow events, and surface hoar.

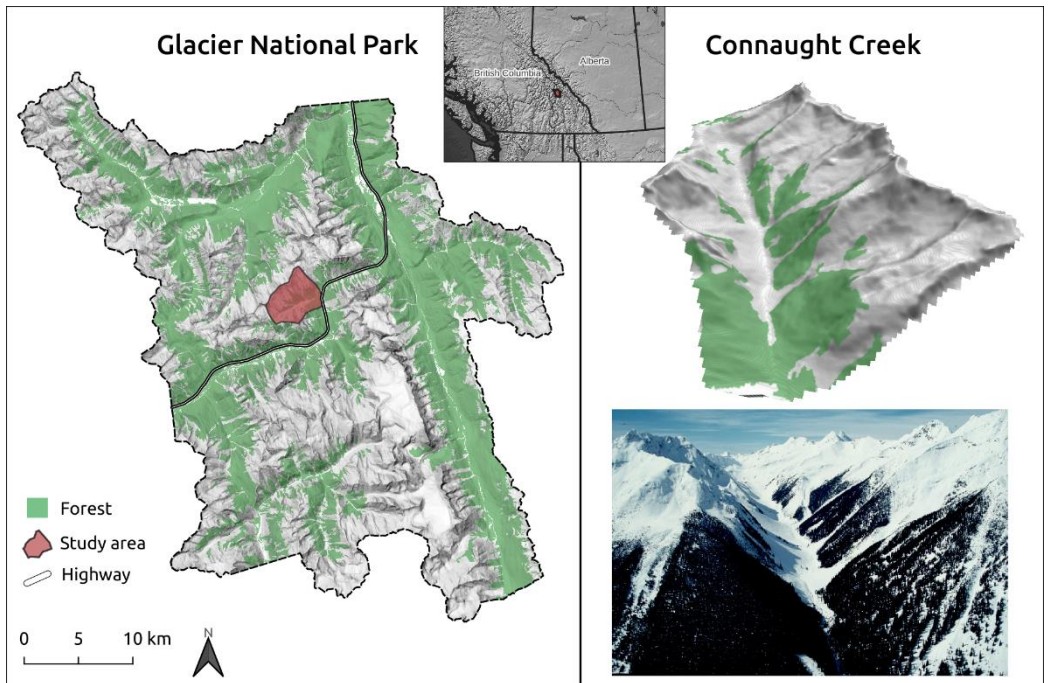

**Figure 1: Connaught Creek study area. Inset map shows location of Glacier National Park. Left panel shows greater area of Glacier National Park, British Columbia, Canada. Right panel shows Connaught Creek in a 3D view and overview photo.**

Rogers Pass has a long-standing avalanche control program due to the presence of the Trans-Canada highway and the Canadian Pacific Railroad. Several high resolution DEM datasets have been generated for this area, which makes it an ideal location to evaluate the impact of DEM resolution and type on the performance of the AutoATES model. Available DEM data include a

1 m LiDAR digital terrain model (DTM), 10 m satellite stereo photogrammetry digital surface model (DSM), 17 m Canadian National DSM, and 26 m Advanced Land Observation Satellite (ALOS) global DSM.

For forest data we used the British Columbia Vegetation Resource Inventory (BC VRI), which provides polygon based forest characteristic information across all of British Columbia, Canada (Sandvoss et al., 2005). The dataset is generated using a combination of aerial photograph interpretation and local study plots to interpolate a wide range of forest characteristics,

including relevant characteristics for capturing avalanche forest interaction such as crown cover, stem density, and basal area (Bebi et al., 2009; Viglietti et al., 2010). We extracted the forest characteristics crown cover, stem density, and basal area from the BC VRI dataset and converted them to raster layers with the same resolution and alignment as the input DEM. We also used the British Columbia Land Cover Classification Scheme to create a binary raster of forest extent based on the BC VRI data set (Ministry of Sustainable Resource Management, 2002).


### 2.2.2 Bow Summit, Banff National Park, AB

Bow Summit (Figure 2) is a popular backcountry recreation site located in Banff National Park in the Rocky Mountains of Alberta, Canada. Elevations range from approximately 1900 m to 2800 m, with a steep alpine ridge running along the western study area boundary which gradually transitions into lower angle forested terrain at lower elevations with several large avalanche paths descending into treeline. Compared to Connaught Creek, this study area does not have overlapping avalanche paths from opposite sides of the valley and therefore the overhead exposure is more well defined.

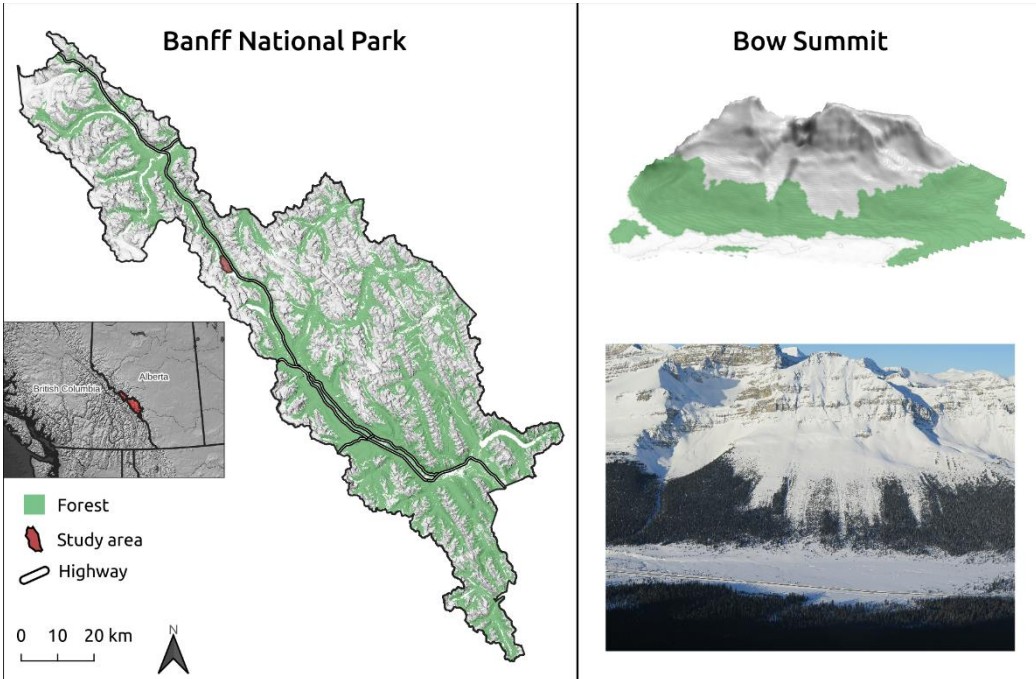

**Figure 2. Bow Summit study area. Inset map shows location of Banff National Park. Left panel shows the greater area of Banff National Park, Alberta, Canada. Right panel shows Bow Summit in a 3D view and overview photo of central portion of study area.**

The snow climate in Bow Summit is continental, which is typical of the Canadian Rocky Mountains and generally characterized by cold temperatures, lower overall snow depth, and presence of persistent weak layers (Haegeli and McClung, 2007). Deep persistent slab avalanche problems are more prevalent in the Canadian Rockies than other mountain ranges in Western Canada, with the most common weak layer types being early-season faceted layers and depth hoar (Shandro and Haegeli, 2018).

There is a notable lack of freely available high resolution DEM data in Bow Summit, with the 26 m ALOS DEM being the best data that we could locate. While the ALOS 26 m DEM is not the optimal input data set for capturing high precision terrain characteristics, it has the benefit of being freely available worldwide and having relatively high accuracy compared to other low resolution global DEMs (Kramm and Hoffmeister, 2019).



For forest data we use the Banff National Park vegetation resource inventory (Parks VRI) which was generated based on the
methods of BC VRI data set. We extracted the same forest variables from the Parks VRI (forest extent, crown cover, stem
density, basal area) to characterize forested areas in Bow Summit.

**2.2 AutoATES**

The basic processing steps of the AutoATES model are: 1. Potential avalanche release area (PRA) modelling. 2. Avalanche
runout simulation. 3. ATES classification (Figure 3). This manuscript is focused on optimization of step 3, the ATES
classification, based on local validation data (Figure 3, step 2-3). This section will provide a brief overview of the entire
AutoATES model as applied in our study areas with a more detailed description of the ATES classification step. See Toft et
al. (2023) for a more detailed description of the processing methods and development of AutoATESv2.0 model.

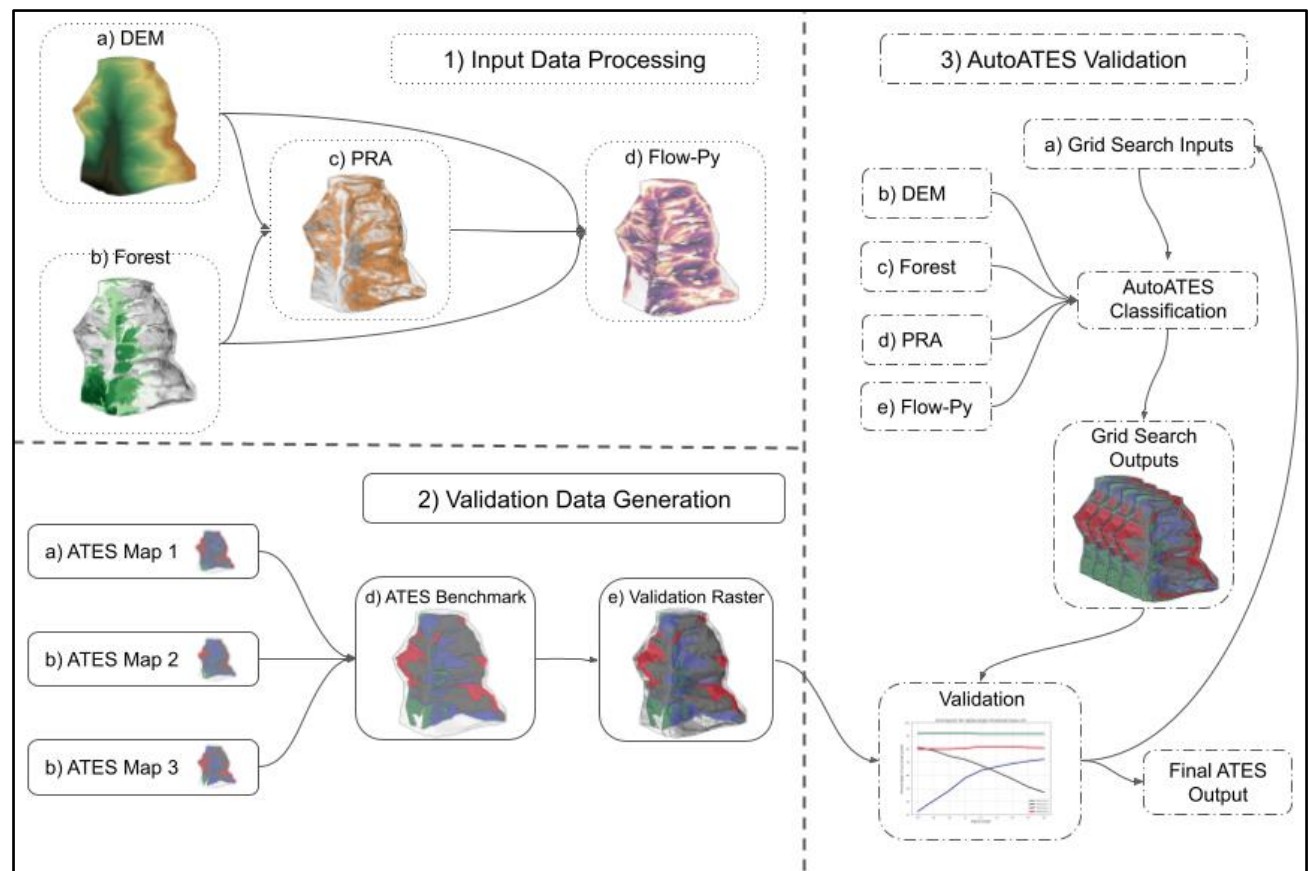

**Figure 3: Automated ATES processing and validation workflow.**





### 2.1.1 Potential avalanche release area model

The PRA model is adapted from Veitinger et al., (2016) with updates to include forest density based on the prior research of Sharp et al. (2018). The output of the PRA model is a continuous raster with values ranging from 0 to 1 indicating the likelihood of each pixel being an avalanche release area. To utilize the PRA output to define the spatial extent of avalanche release areas

in the runout simulation, we created a binary PRA raster using a cut off threshold of 0.15. This PRA cutoff parameter can be tuned based on the desired avalanche frequency scenario and local forest character (Toft et al., 2023). We did not include the PRA threshold in our grid search to optimize the ATES classification because running the subsequent avalanche runout simulations repeatedly requires extensive computer processing time.

### 2.1.2 Avalanche runout simulation

Capturing overhead exposure and avalanche runout in the AutoATES model relies on the Flow-Py avalanche simulation software developed by D'amboise et al. (2022). Based on recommendations from the developers, input from Parks Canada avalanche experts, and local testing we use the following input settings to run Flow-Py: maximum alpha angle 24°, exponent 8, flux threshold 0.003, and max z delta 270. To capture the interaction of forests and avalanche runout, we used the 'forest_detrainment' branch of the Flow-Py GitHub repository. This version of Flow-Py includes functions to account for

forest friction and detrainment of snow in forested terrain (D'amboise et al., 2020). To apply this version of Flow-Py the basal area raster was scaled to the range of 0 to 1 to represent forest density.

### 2.1.3 AutoATES classifier

The final step in the AutoATES modeling chain is to produce the ATES classification. Currently, AutoATES does not include non-avalanche terrain (class 0) due to the high level of certainty required for assigning this class and the associated implications

for risk management decisions. AutoATES classification can be broken down into three distinct steps; 1. Initial classification based on slope angle, alpha angle, and overhead exposure parameters. 2. Updating the ATES classification based on forest density, forest extent, and PRA extent. 3. Removing small islands of cells.

In the first step of the ATES classification, we produce three separate raster layers with ATES ratings solely based on the slope angle, alpha angle, and overhead exposure rasters. The ATES ratings of these layers are determined by the slope angle

thresholds (SAT12, SAT23, SAT34), alpha angle thresholds (AAT12, AAT23), and overhead exposure thresholds (OE12, OE23) that separate the different classes. For each pixel, we take the maximum rating of these three rasters to determine the initial ATES rating (classes 1-4).

Once the initial ATES classification is completed, we use the forest density and PRA extent rasters to update the ATES ratings. Using the initial ATES classification (1-4), forest density (10-40), and the binary PRA (No or Yes) we create a reclassification

scheme with unique values for each combination of ATES rating, forest density, and PRA value (Table 2). Due to low spatial resolution of the BC VRI basal area forest dataset near treeline we used an additional forest dataset, based on the BC Land

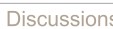

Cover Classification Scheme level 4 vegetation type (Ministry of Sustainable Resource Management, 2002), to create a binary forest extent layer in order to convert pixels that were initially classified as extreme (4) to complex (3). This helps improve the classification accuracy of extreme (4) terrain and aligns with the definition of extreme terrain as having 'few or no trees' in

ATESv2 (Table 2). This step was added to specifically address limitations with VRI basal area data and is optional for other regions depending on the quality of the forest input data.

**Table 2. Forest density reclassification table, italicized text indicates values that are reclassified based on forest density and PRA value (adapted from Toft et al., 2023).**

| | | Initial ATES Rating | | | |
|---|---|---|---|---|---|
| **Forest density** | **PRA Extent** | **Simple (class 1)** | **Challenging (class 2)** | **Complex (class 3)** | **Extreme (class 4)** |
| Open (10) | No (0) | Class 1 | Class 2 | Class 3 | Class 4 |
| | Yes (100) | Class 1 | Class 2 | Class 3 | Class 4 |
| Sparse (20) | No (0) | Class 1 | *Class 1* | *Class 2* | *Class 3* |
| | Yes (100) | Class 1 | *Class 1* | *Class 2* | *Class 3* |
| Moderate (30) | No (0) | Class 1 | *Class 1* | *Class 1* | *Class 3* |
| | Yes (100) | Class 1 | *Class 1* | *Class 2* | *Class 3* |
| Dense (40) | No (0) | Class 1 | *Class 1* | *Class 1* | *Class 2* |
| | Yes (100) | Class 1 | *Class 1* | *Class 2* | *Class 2* |

The final step in producing the ATES classification is to remove isolated islands of pixels smaller than the island filter size.

This helps simplify and smooth the output of AutoATES and bridge the scale mismatch between human and automated maps by removing features that would be unlikely to be included in manual ATES maps. For more details on all AutoATESv2.0 processing steps see Toft et al. (2023).

**2.3 Approach to localization**

Prior versions of AutoATES relied on theoretical thresholds from avalanche literature (Campbell and Gould, 2013; Schweizer

and Lütschg, 2001; McClung and Schaerer, 2023; Lied and Bakkehøi, 1980) to determine the slope angle and alpha angle input parameter values. This approach assumes that real world values captured in the literature are accurately reflected in the DEM data used as input for AutoATES. However, DEMs are a model of real world terrain and therefore offer an imperfect representation of the real world terrain characteristics (Fisher and Tate, 2006). Lower resolution DEM data are especially prone to smoothing terrain features and underestimating slope angles due to the larger area required to calculate derivatives, such as

slope angle and curvature (Hengl and Evans, 2009). For this reason, the direct application of theoretical thresholds from avalanche literature as input parameters for a DEM based model can lead to systematic errors.

As an alternative approach, we used high quality human generated ATES maps to determine the optimal input parameters for automated ATES mapping. This reverse engineering approach uses a grid search method to test a wide range of possible input parameter values and determines the optimal values by quantifying accuracy based on the human ATES maps. The resulting



AutoATES model captures some of the local knowledge and experience of human mappers in determining the optimal settings while having the additional benefit of being transparent, replicable, and efficient for large scale mapping.

### 2.3.1 Reverse engineering with human ATES maps

To generate a validation data set for this research, we collaborated with three avalanche experts from Avalanche Canada and Parks Canada. The three mappers first produced their own independent ATES maps manually, and then collaborated as a group

to reach consensus on the most accurate map. (Figures 4 & 5). For a more complete discussion of the methods for generating manual ATES maps see Statham and Campbell (2023).

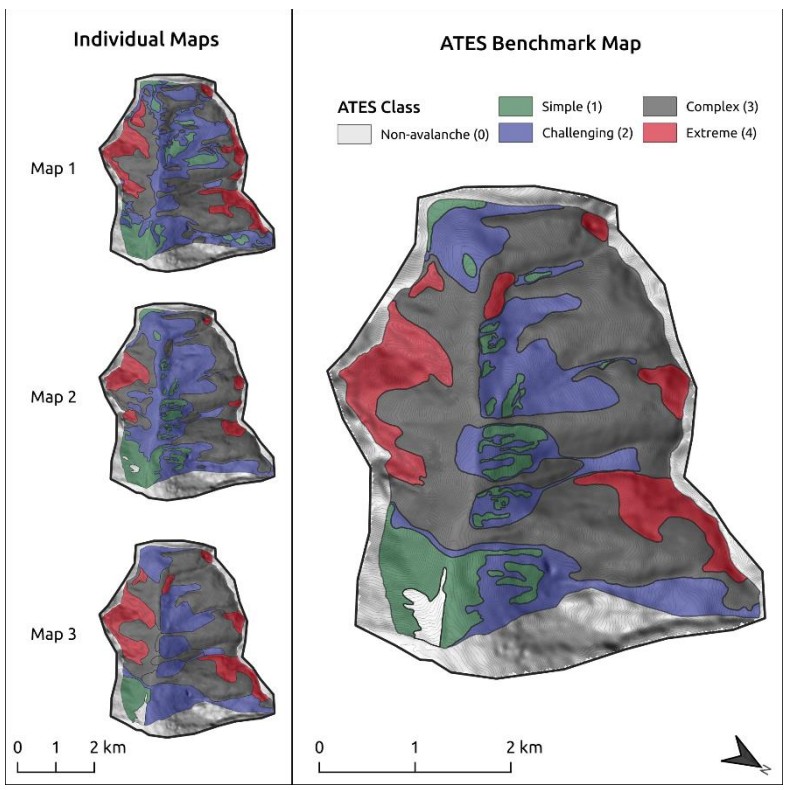

**Figure 4. Connaught Creek study area (12.2 km²) ATES benchmark map and individual human maps used to generate the validation dataset. Note that non-avalanche terrain shown here was considered simple terrain for validation.**

The three individual ATES maps used to generate the benchmark map had substantial differences based on the interpretation of the mapper. These differences highlight the subjectivity in manual ATES mapping and the challenge of having multiple individuals produce consistent ATES maps. The ATES benchmark maps help to reduce subjectivity and provide an ideal validation dataset for localizing and validating the AutoATES model.

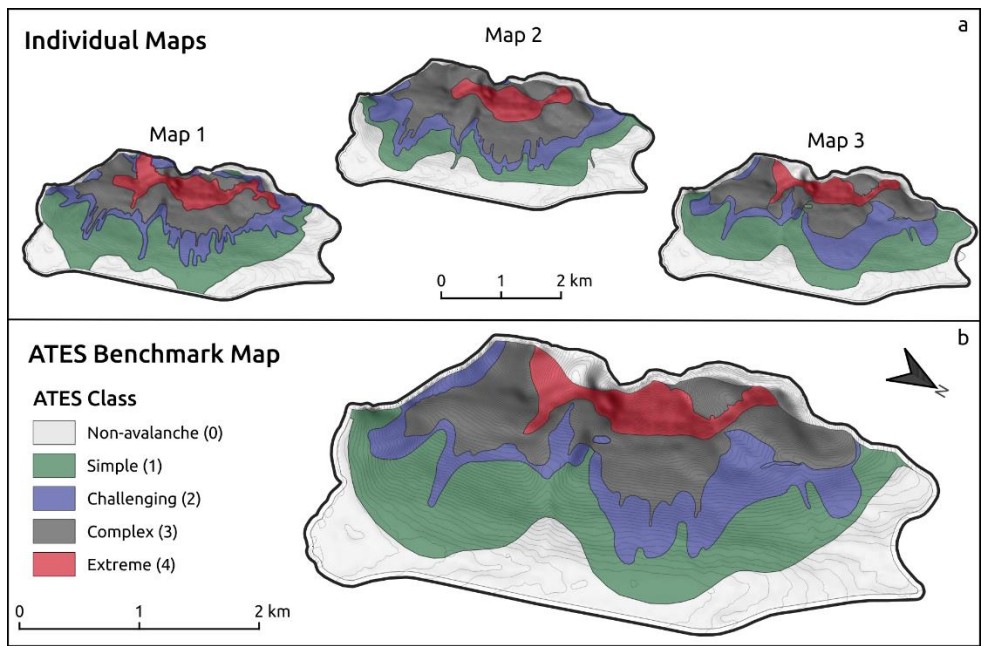

**Figure 5. Bow Summit study area (8.3 km²) ATES benchmark map and individual human maps used to generate the validation dataset. Note that non-avalanche terrain shown here was considered simple terrain for validation.**

### 2.3.2 Grid search

Building on existing avalanche terrain model research (Bühler et al., 2018; Sykes et al., 2022), we used a grid search approach to test a wide range of possible parameter values for AutoATES. The grid search method uses a set of user defined default input parameters as a baseline and iterates over one input parameter at a time based on a list of candidate values for each parameter (Table 4). The baseline parameters are then sequentially updated based on the grid search results until the optimal values are reached (Figure 3). To update the baseline parameters, we started with slope angle thresholds, window size, alpha angle thresholds, forest density thresholds, overhead exposure, then island filter size. After selecting an updated parameter value, we rerun the grid search before moving to the next parameter.

We use a multi-class confusion matrix (Table 3) with the AutoATES output as the predicted values and the rasterized ATES benchmark map as the ground truth values using the python package scikit-learn (Pedregosa et al., 2011) to determine the accuracy of each grid search iteration. Based on the confusion matrix we summarize the accuracy of each grid search iteration based on the percent of all pixels accurately predicted, the percent of pixels accurately predicted for each ATES class, and the percent of all pixels underpredicted (type I errors) and overpredicted (type II errors). We also calculated the macro averaged F1 score, precision, and recall to provide a general comparison of the model performance.

We considered both the accuracy statistics from the confusion matrix and visualized the AutoATES output on a map as a larger scale quality control measure to select the optimal input parameters,. We focus on underestimated pixels instead of overestimated pixels because misclassification into a lower ATES class is a more severe type of error. In cases where the grid





search was inconclusive, we chose the best input parameter based on discussions with local avalanche experts from Parks

Canada and by comparing the input parameter values between our two study areas.

To provide some context for the AutoATES accuracy scores, we compared the three manual ATES maps that were used to create the ATES benchmark map to the final version of the ATES benchmark map. All non-avalanche (0) pixels in the manual maps and benchmark map were converted to simple (1) to facilitate direct comparisons between AutoATES and the human maps, because AutoATES currently does not include non-avalanche (0) terrain. We then used a confusion matrix to calculate

the overall accuracy and the percent of pixels accurately predicted for each ATES class.

**Table 3. Confusion matrix example from grid search accuracy assessment in Bow Summit. Cells below the diagonal indicate underprediction errors (Type I) and cells above the diagonal indicate overprediction errors (Type II).**

|  |  | AutoATES | | | |
| --- | --- | --- | --- | --- | --- |
|  | n = 12462 | Simple (1) | Challenging (2) | Complex (3) | Extreme (4) |
| **ATES Benchmark** | Simple (1) | 6698 (96.6%) | 208 (3.0%) | 25 (0.4%) | 0 |
|  | Challenging (2) | 579 (27.5%) | 1261 (59.9%) | 266 (12.6%) | 0 |
|  | Complex (3) | 6 (0.2%) | 460 (17.9%) | 1947 (75.6%) | 163 (6.3%) |
|  | Extreme (4) | 0 | 1 (0.1%) | 236 (27.8%) | 612 (72.1%) |

## 2.4 DEM sensitivity analysis

The Connaught Creek study area provides an ideal location to examine the effect of different DEM resolutions on our approach

to localize AutoATES because it has a variety of DEM data available. For this research we tested a 5 m LiDAR DEM, 10 m satellite stereo DEM, 17 m national topographic survey-based DEM, and a 26 m free global satellite stereo DEM (Figure 6). The exact resolutions we report are based on the cell sizes after reprojecting the raw geographic coordinate system DEM data into a UTM coordinate system and may vary depending on latitude. We up sampled the LiDAR DEM from 1 m to 5 m for better computational efficiency and based on prior research demonstrating that a 5 m DEM is sufficient for high resolution

avalanche terrain modeling (Bühler et al., 2011, 2013).

Because the extent of the LiDAR DEM did not cover the entire area of the Connaught Creek ATES benchmark map, we clipped an area of 0.14 km$^2$ from each of the DEM datasets to provide a consistent comparison. We adjusted the windshelter radius input parameter of the PRA model based on the resolution of each DEM and otherwise used the same input parameters for the PRA model and Flow-Py for all DEM datasets (Toft et al., 2023). To determine the optimal input parameters for each

DEM, we applied the same grid search process described previously.

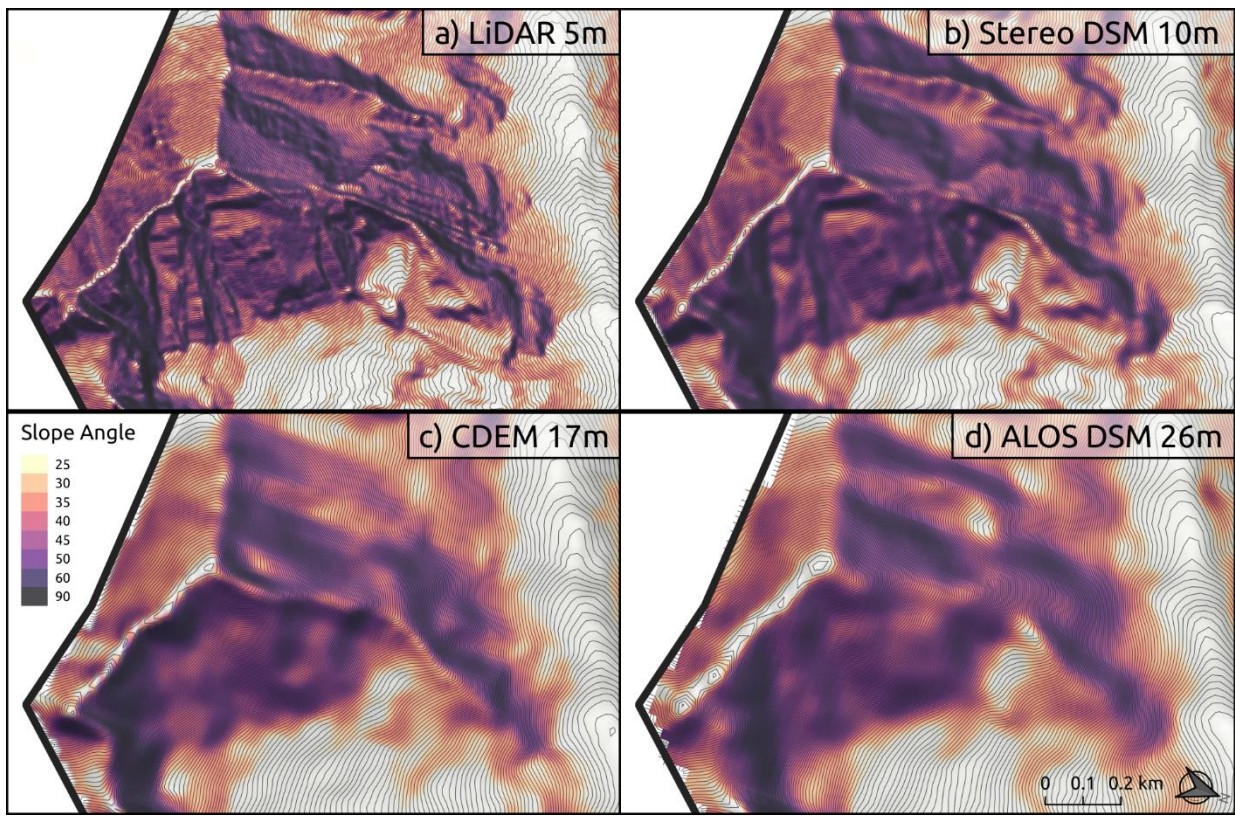

**Figure 6. Slope angle rasters from the LiDAR DEM (a), satellite stereo DSM (b), Canadian national DEM (c), and ALOS DSM (d). The wide variety of DEM data illustrate the smoothing effect of lower resolution input data.**

## 3 Results

### 3.1 Localization of parameters

To determine the optimal input parameters for AutoATES, we performed a grid search using the ALOS 26 m satellite stereo DEM, which is the best available DEM that covers both our Connaught Creek and Bow Summit study areas. Out of the 12 AutoATES input parameters 5 had identical values between Bow Summit and Connaught Creek (AAT12, OE23, TREE12, TREE23, ISL_SIZE) (Table 4). Of the remaining 7 parameters, 6 had values that differed by only one step in our grid search. The only parameter with a difference greater than 1 step on the grid search is the overhead exposure threshold between challenging (2) and complex (3) terrain (OE23).





**Table 4. Optimized AutoATES input parameters based on the ALOS DEM for the Bow Summit and Connaught Creek study areas.**

| Input Raster | Input Parameter | Range Tested | Iterations | Connaught Creek | Bow Summit |
|---|---|---|---|---|---|
| Slope angle threshold | SAT12 | 15° - 30° | 16 | 18° | 19° |
| | SAT23 | 30° - 40° | 11 | 29° | 28° |
| | SAT34 | 35° - 50° | 16 | 39° | 40° |
| | WIN_SIZE | 1 – 50 | 18 | 3 | 1 |
| Alpha angle threshold | AAT12 | 20° - 28° | 9 | 24° | 24° |
| | AAT23 | 28° - 36° | 9 | 34° | 33° |
| Overhead exposure | OE12 | 0 – 50 | 13 | 5 | 5 |
| | OE23 | 10 – 75 | 14 | 25 | 40 |
| Forest density | TREE01 | 5 – 50 | 10 | 5 | 10 |
| | TREE12 | 10 – 65 | 12 | 20 | 20 |
| | TREE23 | 25 – 80 | 12 | 25 | 25 |
| Island filter size | ISL_SIZE | 5000 – 50000 m$^2$ | 10 | 30000 m$^2$ | 30000 m$^2$ |

### 3.1.1 Slope angle

Adjusting the slope angle parameters (SAT12, SAT23, SAT34, WIN_SIZE) has the largest overall impact on the output of AutoATES as indicated by the charts in the panels in the top row of Figures 8 and 9. For the slope angle threshold between simple (1) and challenging (2) (SAT12) we chose to limit the optimized value to 18° despite the grid search indicating that a lower value would improve the accuracy of challenging (2) terrain (Figures 8 & 9). We made this choice because simple (1) terrain is defined as having slope angles less than 20° with steeper areas of dense forest (Statham and Campbell, 2023), but allowing AutoATES to decrease the value to 18° allows us to account for the effects of DEM smoothing. Lower SAT12 values causes non-forested alpine terrain with low avalanche exposure to be systematically overpredicted by AutoATES. We believe this type of terrain is underrepresented in our two study areas and therefore not illustrated in our grid search results.

Our selection of the SAT23 and SAT34 parameters was primarily driven by slight spikes in overall accuracy for the respective values of Bow Summit and Connaught Creek. We also preferred parameter values that increased the accuracy of challenging (2) terrain, since it has by far the lowest accuracy of the ATES classes. The goal of including the WIN_SIZE input parameter is to filter out small steep terrain features that may have locally steep slope angles, but the overall terrain feature is not large enough to constitute an extreme (4) rating. For Connaught Creek there was a notable increase in overall accuracy when WIN_SIZE was set to 3 due to subtle increases in accuracy for simple (1), challenging (2), and complex (3) despite a decrease in accuracy for extreme (4).

### 3.1.2 Alpha angle

The impact of alpha angle on the output of AutoATES is less significant than slope angle, which can be seen in the lack of variation in overall accuracy and ATES class accuracy across the range of grid search values for AAT12 and AAT23 (Figure





7, 8). This is especially true for AAT12, which has a minor impact on accuracy of challenging (2) terrain for Connaught Creek

and Bow Summit. This behaviour is an artifact of our grid search procedure only varying one input parameter at a time and keeping all others constant. For challenging (2) terrain the spatial extent is similar for alpha angle and overhead exposure, therefore the impact of changes in the alpha angle threshold are muted by the overhead exposure parameters (OE12, OE23) remaining the same.

The alpha angle threshold between challenging (2) and complex (3) (AAT23) terrain has more variation across the range of

grid search values (Figures 7 & 8). In both Connaught Creek and Bow Summit we set AAT23 to improve the accuracy of challenging (2) terrain without significant negative impacts on the overall accuracy or percent of underpredicted pixels. However, there is a proportional decrease in the accuracy of complex (3) terrain.

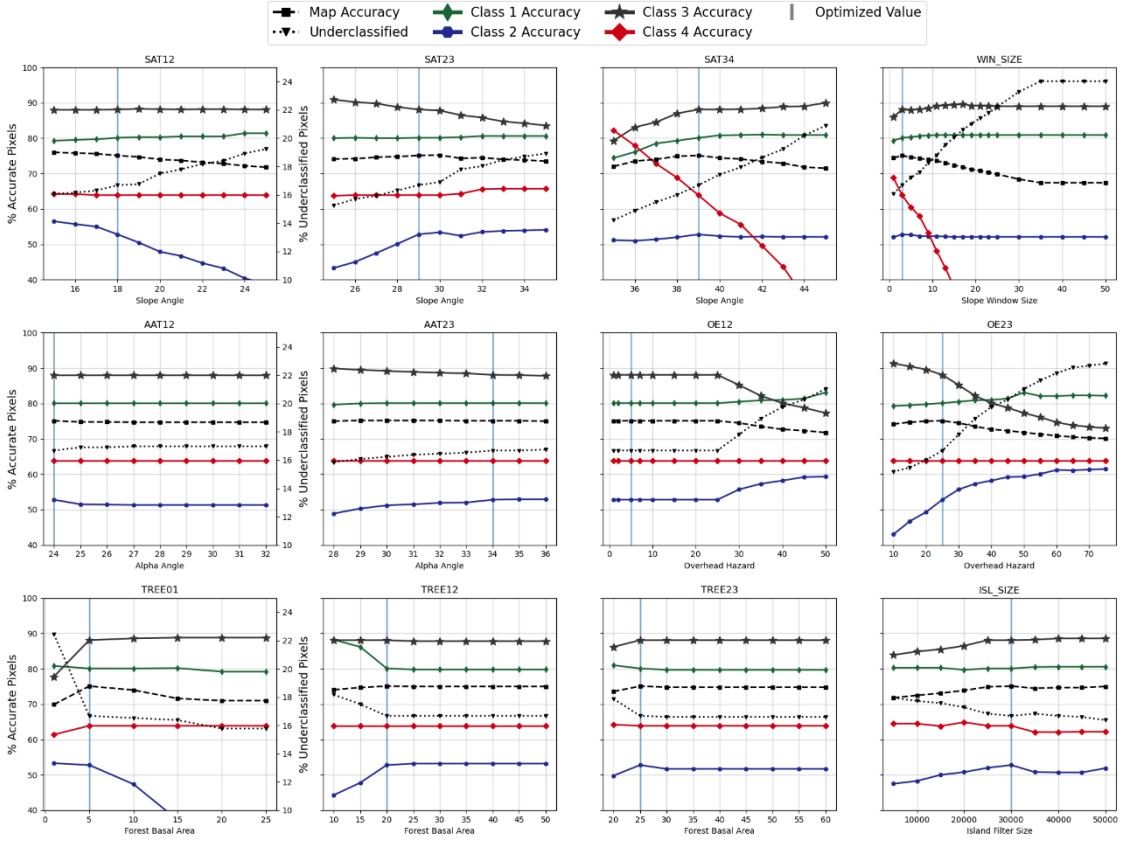

**Figure 7. Results of the grid search procedure for the Connaught Creek study area. To select optimal parameter values, we**
**considered overall accuracy, percent underpredicted, and balanced accuracy across ATES classes.**


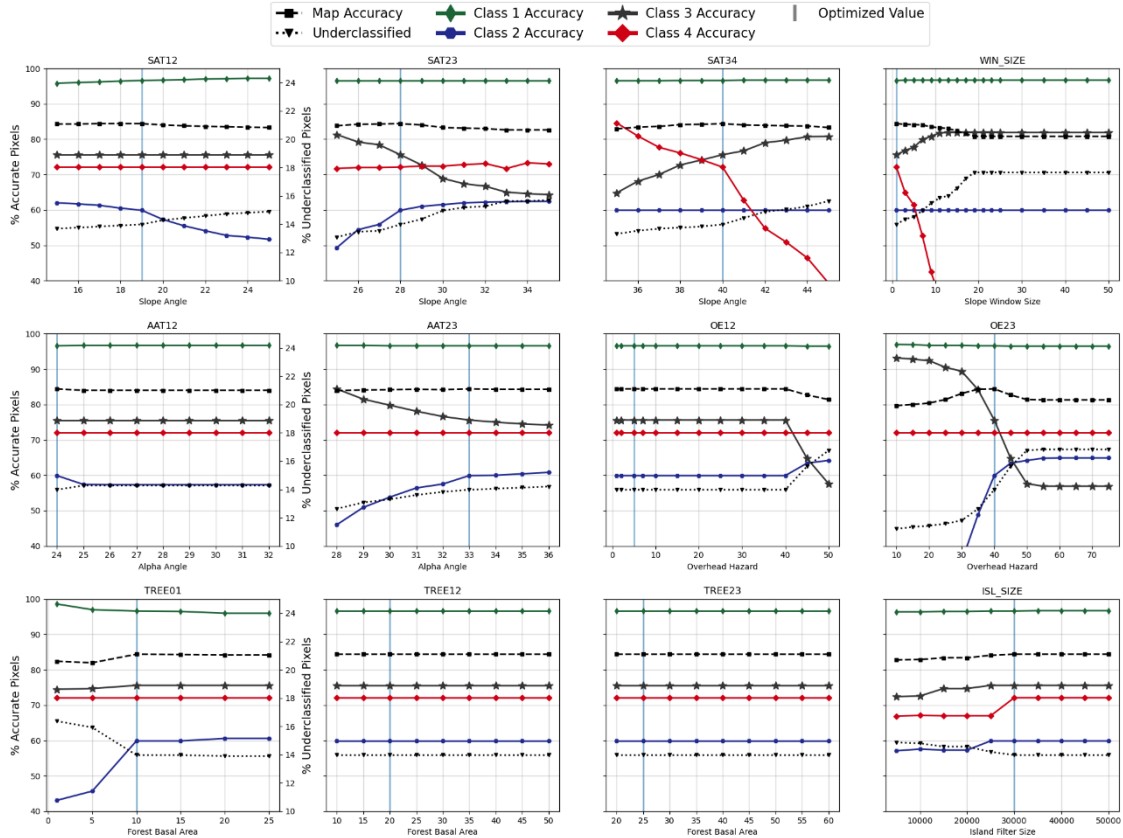

**Figure 8. Results of the grid search procedure for the Bow Summit study area. To select optimal parameter values, we considered overall accuracy, percent underpredicted, and balanced accuracy across ATES classes.**

### 3.1.3 Overhead exposure

Based on the grid search we see no variation in overall accuracy, ATES class accuracy, or percent of underpredicted pixels to determine the optimal threshold between simple (1) and challenging (2) terrain for overhead exposure (OE12). Therefore, we turned to the definitions of ATESv2 for guidance (Statham and Campbell, 2023). Simple (1) terrain is defined as having runout zones with well-defined path boundaries where deposits fan out, crossing low frequency runout zones, and minimal exposure crossing runouts. Therefore, we decided to set OE12 to 5 out of a possible maximum of 100. This means that overhead exposure

could exist but the average flow velocity and number of overhead start zones that could impact a given pixel are only 5 % of the possible maximum for a given study area.

For the overhead exposure threshold between challenging (2) and complex (3) terrain the grid search illustrates an increase in overall accuracy at the respective values for each study area (Figures 7 & 8). We also used this parameter as another tool to improve accuracy of challenging (2) terrain but were conscious of the coincident sharp increase in underpredicted pixels which

prevented us from increasing the value any further for both study areas.





### 3.1.4 Forest density thresholds

The grid search provided more definitive results to set the optimal input parameters for forest density in Connaught Creek versus Bow Summit. We see higher values of overall accuracy and increased accuracy in challenging (2) terrain at each of the optimized parameters of TREE01, TREE12, and TREE23 in Connaught Creek. For Bow Summit we altered the value of
TREE01 to 10 m$^2$ per hectare based on higher overall accuracy and challenging (2) terrain accuracy. The overall lack of variability in the TREE12 and TREE23 parameters is mostly due to the spatial resolution of the input dataset. The forest data was generated as polygons delineated from aerial photograph interpretation with single values for each attribute assigned for each polygon. This results in a raster layer with limited small scale variability.

### 3.1.5 Island filter size

Filtering out groups of pixels below the island filter size is beneficial for improving the accuracy of challenging (2) and complex (3) terrain in both study areas. In Bow Summit, it is also beneficial for improving the accuracy of extreme (4) terrain. In both study areas there is a negligible effect on simple (1) terrain. This is due to the fact that most of the small polygons in the ATES Benchmark map are in simple (1) terrain, which are delineated by small clusters of dense forest. The fact that the island filter improves the accuracy of AutoATES is likely due to our use of manual ATES maps as a validation dataset. It is
not practical, or the intent of the ATESv2.0 definitions, to evaluate terrain characteristics on a pixel-by-pixel basis when generating manual ATES maps and therefore a certain degree of smoothing is implicit to the manual mapping process.

### 3.2 Accuracy of localized models

The overall accuracy of AutoATES compared to the ATES benchmark maps is 75.1 % for Connaught Creek (Figure 9, Table 7) and 84.4 % for Bow Summit (Figure 10, Table 7) using the ALOS 26 m DEM for both study areas. There is substantial
variation in accuracy across the ATES classes for both study areas. Simple (1) terrain has the highest accuracy rating in Bow Summit (96.6 %) and the second highest accuracy rating for Connaught Creek (80.0 %) (Table 5). In Connaught Creek 20 % of simple (1) terrain is overpredicted by AutoATES, with 15.5 % of those pixels predicted as challenging (2) and 4.5 % predicted as complex (3). The 4.5 % of pixels overpredicted by 2 classes is primarily due to overestimating runout distance and overhead exposure near the mouth of Connaught Creek at the bottom of the map in Figure 9.
Challenging (2) terrain has the lowest accuracy in both study areas (Table 5) and a relatively large proportion of underpredicted pixels. Examples of underprediction due to forest extent exist in the central portion of the Connaught Creek map, where the patches of simple (1) terrain in the AutoATES map are much larger than the ATES benchmark map, and in the estimation of treeline in the center right portion of the Bow Summit AutoATES map, where simple (1) terrain extends too far upslope compared to the ATES benchmark map. In addition, simple terrain was identified in alpine areas of Connaught Creek, such as
the bottom right of the AutoATES map, which was not captured by the ATES benchmark maps.

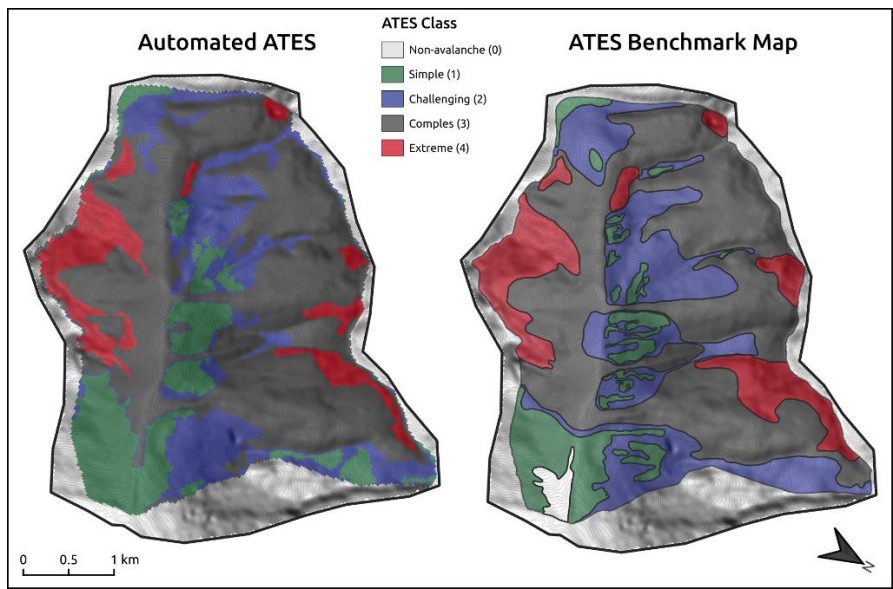

**Figure 9. Automated ATES output from grid search procedure in Connaught Creek study area (left panel) and ATES benchmark map used for validation (right panel).**


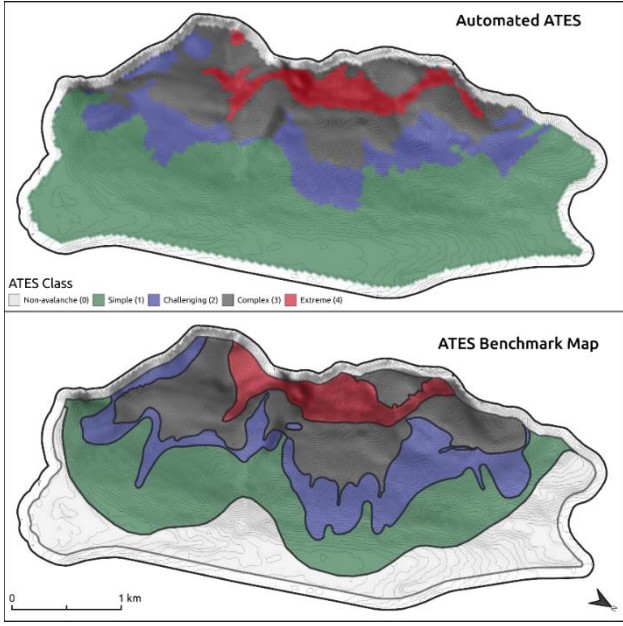

**Figure 10. Automated ATES output from grid search procedure in Bow Summit study area (top panel) and ATES benchmark map used for validation (bottom panel).**






**Table 5. AutoATES accuracy and percentage of under and over-classified pixels by ATES class compared to ATES benchmark maps**

| ATES Class | | Bow Summit | Connaught Creek |
|---|---|---|---|
| Simple (1) | Accuracy | 96.6 | 80.0 |
| | Under | *NA* | *NA* |
| | Over | 3.4 | 20.0 |
| Challenging (2) | Accuracy | 59.9 | 52.8 |
| | Under | 27.5 | 20.6 |
| | Over | 12.6 | 26.6 |
| Complex (3) | Accuracy | 75.6 | 88.1 |
| | Under | 18.1 | 8.3 |
| | Over | 6.3 | 3.6 |
| Extreme (4) | Accuracy | 72.1 | 63.9 |
| | Under | 27.9 | 36.1 |
| | Over | *NA* | *NA* |

Complex (3) terrain is the second most accurate class in Bow Summit and the most accurate in Connaught Creek. The rate of underpredicted pixels is much higher in Bow Summit, with 18.1 % of complex (3) terrain being predicted as challenging (2) (17.9 %) or simple (1) (0.2 %). The majority of these underpredicted pixels are due to differences in the estimation of runout or overhead exposure. For example, in the left center portion of Figure 10 where two lobes of complex terrain extend further downslope in the ATES benchmark map.

Extreme (4) terrain has the second worst accuracy for both Connaught Creek and Bow Summit and the highest rate of underpredicted pixels for both study areas. In both cases the majority of the underestimated area is localized to a single portion of the map which has relatively lower slope angle values compared to the rest of the extreme (4) terrain. In Connaught Creek the underpredicted area is in the lower right of the ATES benchmark map where extreme terrain extends much further downslope (Figure 9). In Bow Summit there is an area of extreme (4) terrain in the top center of the ATES benchmark map that is connected to the large cliff section below which is not connected in the AutoATES map.

### 3.2.1 Comparison to manual ATES maps

To get a sense of the overall skill of the AutoATES model, we compare the accuracy metrics to the performance of the three manual ATES maps used to generate the ATES benchmark map. For both Connaught Creek and Bow Summit the overall accuracy, precision, recall, and F1 macro of AutoATES falls within the range of the three manual maps (Table 6). The average overall accuracy of the manual maps is 84.8 % for Bow Summit and 76.1 % for Connaught Creek compared to 84.4 % and 75.1 % for AutoATES.




**Table 6. Comparison of AutoATES to manual ATES maps.**

| ATES Map | Study Area | Overall Accuracy | Precision | Recall | F1 Macro |
|---|---|---|---|---|---|
| AutoATES | Bow Summit | 84.4 | 79.0 | 76.1 | 77.5 |
| | Connaught Creek | 75.1 | 73.9 | 71.0 | 71.6 |
| Map 1 | Bow Summit | 80.7 | 70.9 | 76.3 | 72.7 |
| | Connaught Creek | 67.9 | 66.3 | 66.7 | 66.1 |
| Map 2 | Bow Summit | 76.6 | 66.7 | 70.0 | 67.8 |
| | Connaught Creek | 74.3 | 81.3 | 76.8 | 76.2 |
| Map 3 | Bow Summit | 97.1 | 96.7 | 97.1 | 96.8 |
| | Connaught Creek | 86.0 | 87.1 | 80.5 | 80.4 |

Taking a closer look at accuracy scores by ATES class, we can see that in simple (1) terrain the manual maps and AutoATES performed better in Bow Summit compared to Connaught Creek (Figure 11). In challenging (2) terrain, AutoATES was relatively consistent in the two study areas while the manual maps had higher average accuracy in Connaught Creek versus Bow Summit. In complex (3) terrain, AutoATES performed better in Connaught Creek while the manual maps have a similar average accuracy between the two study areas. In extreme (4) terrain, the manual maps generally performed better than AutoATES with higher average accuracy in both study areas.

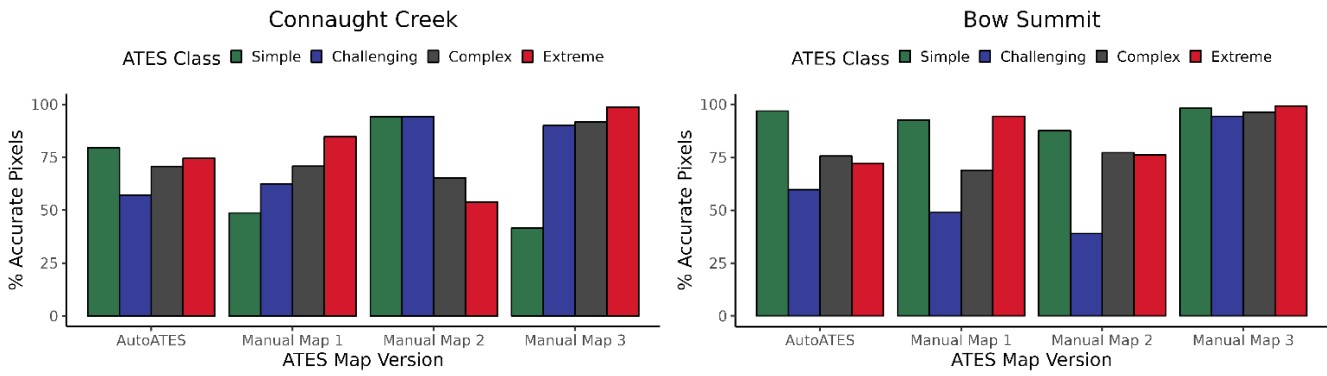

**Figure 11. Performance of AutoATES and three human mappers compared to ATES benchmark map for Connaught Creek (left) Bow Summit (right) color coded by ATES class simple (green), challenging (blue), complex (black), and extreme (red) terrain**

### 3.3 Effect of DEM resolution

Our DEM sensitivity analysis revealed that the resolution and type of input DEM does not have a large impact on the overall accuracy of the AutoATES model (Figure 12). The overall accuracies are all within 1.2 percentage points of one another (Table 7), which is a negligible difference given the dramatic variation in accuracy and precision between the DEM inputs.



The grid search plots for each DEM, which illustrate the change in accuracy across a range of possible input values, are included in appendix A.

**Table 7. DEM sensitivity analysis results**

| DEM Input | Overall | Simple (1) | Challenging (2) | Complex (3) | Extreme (4) |
|---|---|---|---|---|---|
| ALOS 26 m | 75.1 | 80.0 | 52.8 | 88.1 | 63.9 |
| CDEM 17 m | 74.8 | 81.2 | 53.2 | 85.7 | 68.6 |
| Stereo 10 m | 73.9 | 78.4 | 53.5 | 82.0 | 78.3 |
| LiDAR 5 m | 74.1 | 80.7 | 53.3 | 82.4 | 76.2 |

The accuracy scores are very similar between all DEM input datasets for simple (1) and challenging (2) terrain (Table 7). This is at least partially due to the forest input data remaining the same across all DEM versions and therefore limiting the impact of higher resolution terrain data in forested areas. The lowest resolution ALOS 26 m DEM had the highest accuracy for complex (3) terrain, 2.4 % - 6.1 % higher than the other DEM input data. Differences in the lateral spreading of the runout

model due to DEM resolution and lower rates of overprediction in extreme (4) terrain are mostly responsible for the improved performance in complex (3) terrain. The higher resolution DEM data generally performed better in extreme (4) terrain with increases of 4.7 % - 14.4 % compared to the ALOS 26 m DEM. However, since there is less total area of extreme (4) terrain in the ATES benchmark map and the increased accuracy is accompanied by decreased accuracy in complex (3) terrain the overall accuracy is still slightly lower.

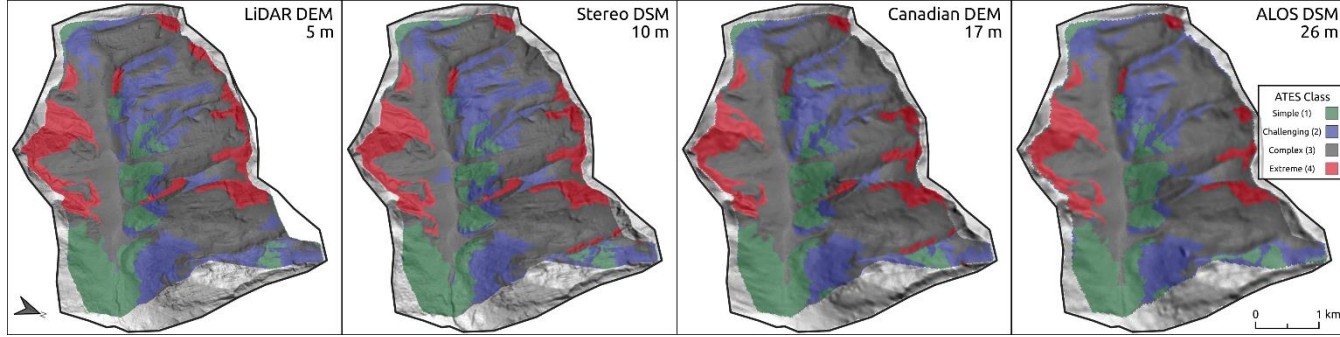


**Figure 12. DEM sensitivity analysis for Connaught Creek study area, showing the spatial patterns between the output have some variation but are largely consistent despite vastly different accuracy of the input DEM data.**

The main differences in the optimized input parameter values across the DEM datasets are in the slope angle thresholds and overhead exposure threshold (OE23) (Table 8). For the slope angle threshold between challenging (2) and complex (3) the

LiDAR DEM has a higher threshold compared to the other DEM data, which can be attributed to a lesser degree of terrain smoothing in the higher resolution data. Differences in the slope angle threshold between complex (3) and extreme (4) terrain are more difficult to compare directly due to the impact of the increased window size that is used to apply a smoothing filter





to the slope angle layer before the extreme (4) slope angle threshold is applied. We determined the optimized values of WIN_SIZE for each DEM based on higher values of overall accuracy in the grid search plots (Appendix A) which are caused

by slight increases in accuracy of simple (1) and complex (3) terrain. It is logical that the window size would increase as raster resolution increases to maintain a relatively consistent filtering effect on small terrain features. The actual window sizes, accounting for DEM resolution, are approximately 6000 $m^2$ for the ALOS 26 m DEM, 2600 $m^2$ for the CDEM, 17000 $m^2$ for the Stereo DEM, and 22500 $m^2$ for the LiDAR DEM.

**Table 8. AutoATES localized input parameters for Connaught Creek by DEM dataset.**

| Input Raster | Input Parameter | ALOS 26 m | CDEM 17 m | Stereo 10 m | LiDAR 5 m |
|---|---|---|---|---|---|
| Slope angle threshold | SAT12 | 18° | 18° | 19° | 18° |
| | SAT23 | 29° | 28° | 29° | 31° |
| | SAT34 | 39° | 38° | 37° | 37° |
| | WIN_SIZE | 3 | 3 | 13 | 30 |
| Alpha angle threshold | AAT12 | 24° | 24° | 24° | 24° |
| | AAT23 | 34° | 34° | 34° | 34° |
| Overhead exposure | OE12 | 5 | 5 | 5 | 5 |
| | OE23 | 25 | 30 | 35 | 30 |
| Forest density | TREE01 | 5 | 5 | 5 | 5 |
| | TREE12 | 20 | 20 | 20 | 20 |
| | TREE23 | 25 | 25 | 25 | 25 |
| Island filter size | ISL_SIZE | 30000 | 30000 | 30000 | 35000 |

## 4 Discussion

### 4.1 Evaluation of localized models

Our primary research objective is to evaluate the accuracy of the AutoATES model against the standard practices of ATES mapping in Western Canada. Our reverse engineering approach produced overall accuracies of 75.1 % in Connaught Creek

and 84.4 % in Bow Summit. In comparison, three manually generated ATES maps created by avalanche experts have an average accuracy of 76.1 % in Connaught Creek and 84.8 % in Bow Summit. The similarity of the overall accuracy scores as well as precision, recall, and F1 macro performance metrics (Table 6) demonstrate that the localized versions of AutoATES have comparable skill to human mappers in generating ATES ratings. For a more detailed description of how the development of AutoATESv2.0 has contributed to improved accuracy compared to the original version of AutoATES see Toft et al., 2023.





The large difference in accuracy scores between the two study areas is partially due to the added complexity of the terrain in Connaught Creek, with many smaller drainages and bowls joining together into the main valley. This makes the assessment of overhead exposure and delineation of boundaries between ATES classes more challenging. Another contributing factor is the relatively large number of small simple (1) polygons in the Connaught Creek ATES benchmark map that are delineated based on local knowledge of forest characteristics (Figure 9). The resolution of the VRI forest data used in this research is not

sufficient to capture these features and in some cases the local knowledge captured in the ATES benchmark map differs from the basal area values. The VRI forest data are generated based on a combination of forest sample plots and manual interpretation of aerial imagery (Sandvoss et al., 2005). While these methods are efficient for large scale mapping and gross estimation of forest characteristics, they do not have adequate resolution in the upper treeline elevations to accurately capture the interaction of avalanches and forest.

Accuracy for each ATES class varies by study area but overall, challenging (2) terrain has the lowest accuracy and relatively high rates of underpredicted pixels (type I error). In the Bow Summit study area, the individual manual ATES maps also consistently had the lowest accuracy in challenging (2) terrain, indicating that it is difficult to consistently classify for human ATES mappers as well. Underprediction errors are the most consequential for AutoATES due to end users' interpretation that the terrain is safer than what has been determined in the ATES benchmark map. This is especially important for the delineation

of simple (1) and challenging (2) terrain because of the risk management implications of how simple terrain is interpreted and applied by backcountry recreationists. The majority of underpredicted pixels in challenging (2) terrain in both study areas are due to the relatively low resolution and precision of the BC VRI and Parks VRI forest datasets.

Extreme (4) terrain is another area where AutoATES has high rates of underpredicted pixels and could be substantially improved. Currently slope angle and forest extent are the only parameters used to delineate extreme (4) terrain. One critical

element that is not currently captured in AutoATES is terrain exposure. The consequences of falling or being pushed off a steep slope by a small avalanche are critical considerations for extreme (4) terrain which are not captured by the PRA or Flow-Py models.

### 4.1.1 Differences in localized parameters by study area

Overall, the localized input parameter settings between Bow Summit and Connaught Creek are similar when using the ALOS

26 m DEM. Out of 12 input parameters, 5 have the same optimized values for both study areas, 6 differ by only one step in the grid search method, and 1 differs by more than one step in the grid search. For the parameters that differ between the two study areas the differences are mostly driven by the goal of improving performance in challenging (2) terrain. The two input parameters with the most impact on the accuracy of challenging (2) terrain are OE23 and TREE01 (Figures 7 & 8).

For Bow Summit, lowering the OE23 value to match Connaught Creek would result in a steep decline in accuracy of

challenging (2) terrain and a decrease in overall accuracy of roughly 4 % (Figure 7). For Connaught Creek, increasing the value of OE23 to match Bow Summit would result in a dramatic increase in the rate of underpredicted pixels (Figure 8). These localized differences in parameter settings are due to multiple factors including differences in the range of possible overhead





exposure values between the two study areas and the potential for slight differences in how the ATES benchmark maps classify challenging terrain in the two study areas.

Overhead exposure is calculated as a function of the average of the z_delta and cell_counts output of the Flow-Py avalanche runout simulation software (Toft et al., 2023). The z_delta layer has a fixed maximum value of 270 which is defined as an input parameter of Flow-Py (D'amboise et al., 2022). However, the cell_counts layer represents the number of release area pixels that could potentially impact a given runout area cell, which can have a wide range of maximum values in areas with multiple large channelized overlapping avalanche paths. For example, the maximum cell_counts value using the ALOS 26 m

DEM for Connaught Creek is roughly 4000 versus 500 in Bow Summit. In the interest of making the overhead exposure layer more standardized across different types of terrain and with different DEM resolutions we take the natural log of the cell_counts layer and scale it the range of 0 to 100 before combining it with the z_delta layer. However, the resulting overhead exposure layer can still vary considerably depending on the maximum cell_counts value for a given study area.

The forest density parameter for determining the threshold between open and sparse forest (TREE01) is also important for

accurately capturing challenging (2) terrain. This is primarily due to how the ATES rating is adjusted in forested areas after the initial ATES classification step. The basal area threshold to delineate sparse forest is 5 m$^2$ per hectare in Bow Summit and 10 m$^2$ per hectare in Connaught Creek. It is beyond the scope of this manuscript to conduct field site visits to determine whether these parameter differences are due to actual differences in forest character on the ground or due to the variability in the VRI datasets. The Connaught Creek area receives more annual precipitation compared to Bow Summit (Shandro and Haegeli, 2018;

Haegeli and McClung, 2007) which could lead to generally denser vegetation. To apply AutoATES in other areas, we recommend users carefully consider the forest data available and ecological climate to accurately set the forest density parameters.

The relative consistency of the localized slope angle and alpha angle input parameters is an indication that there are not substantial differences in the slope angle distribution for avalanche release or the runout distance of avalanches between our

two study areas (McClung and Mears, 1991; Jamieson et al., 2018). This is likely due to the fact that we are using low resolution DEM input data and the fact that persistent weak layers are common in both locations. While the common weak layer types and overall snowpack character differ between the two study areas, the shared propensity for development of persistent weak layers and the associated terrain management considerations could lead to the input parameters values being largely similar.

### 4.2 Effect of DEM type and resolution

Our results show that DEM type and resolution have minimal impact on the overall accuracy of AutoATES in Connaught Creek. Using four different DEM datasets ranging from 5 m to 26 m resolution we performed a grid search to optimize the input parameters and found that all DEMs produce ATES ratings with accuracies of roughly 74.5 %, ± 1 % (Table 8). The fact that lower resolution DEM data performs equally as well as high resolution DEM data is an advantage for processing AutoATES on large scales and makes the application of the AutoATES model more feasible in different regions worldwide.





Prior research has clearly demonstrated that the accuracy of PRA models and runout simulations are sensitive to DEM type and resolution (Brožová et al., 2021, 2020; Bühler et al., 2011, 2013). Therefore, the accuracy of the individual components of the AutoATES model, which include a PRA model and runout model, are assumed to be more accurate with higher resolution DEM data. The greater accuracy of high resolution DEM data is also clearly shown by the increased precision in identifying accurate slope angles and terrain features relevant for avalanche release (Figure 6).

One notable difference between the DEM types is the amount of lateral spreading that occurs in channelized terrain features in the Flow-Py runout simulations. For the CDEM and ALOS DEM data, the width of the avalanche runout zones in gully features extends towards the visible trim lines of forested terrain (Figure 13 c-d) indicating that these areas are impacted by avalanches frequently enough to deter growth of forests (McClung and Schaerer, 2023). However, in LiDAR and Stereo DEM data the lateral spreading of the runout in gully features is limited to the width of the incised creek bed (Figure 13 a-b). Prior

research has demonstrated that terrain smoothing due to snow loading has a strong impact on avalanche release and runout modeling (Veitinger et al., 2014). Based on this observation we recommend exploring DEM smoothing functions based on snow depth, resampling the DEM to a lower resolution, or adjusting the input parameters of Flow-Py for more realistic runout modeling when using higher resolution DEM data.

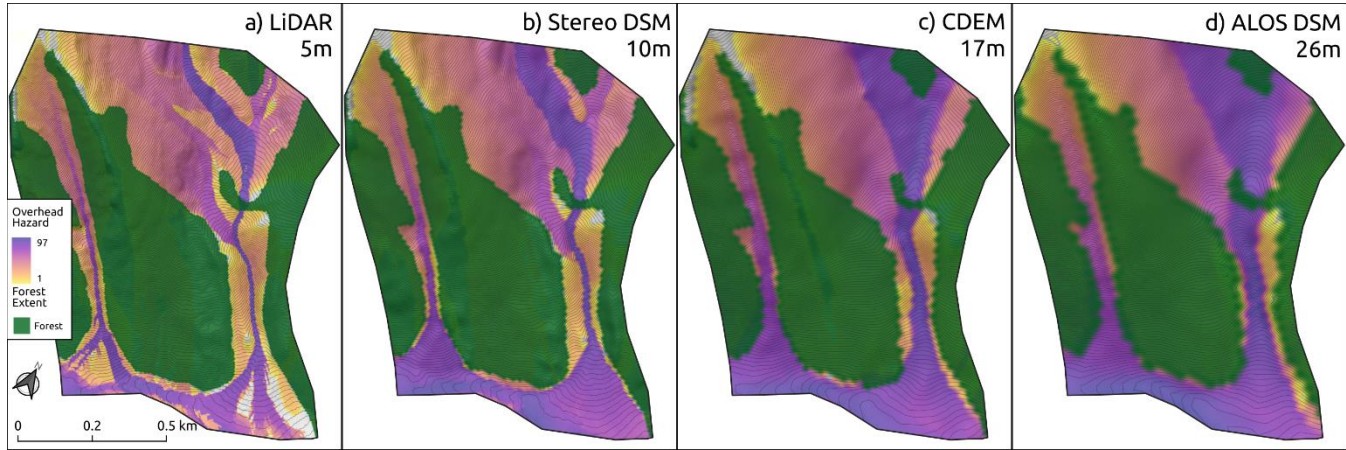


**Figure 13. Effect of DEM resolution on runout simulation in Flow-Py in channelized terrain features. Note the difference in the width of the high overhead exposure region in the two gullies and the relationship to the extent of forested terrain.**

In addition, the Stereo 10 m DEM has areas with unrealistically steep terrain compared to the other DEM datasets along the margins of forested terrain. This is a common issue with Stereo DEMs, because they are a digital surface model (DSM) which

represent the first reflective surface (i.e. top of forest canopy). These errors do not appear to have a major impact on the overall accuracy for the Stereo DEM but are notable in slope angle layers and PRA output and should be considered when using Stereo DEM data as input to AutoATES.



### 4.2.1 Simplicity and scale of ATES system

There are two likely high-level causes of the DEM input having no effect on the accuracy of AutoATES. First is the fact that

the ATES system is designed around generalizing terrain features into broad categories to simplify end user interpretation of terrain exposure. The nature of the ATES system is to de-emphasize precise measurements of any individual terrain characteristic and focus on a more holistic view of multiple terrain characteristics. Based on that view of the terrain, automated ATES models would benefit more from developing meaningful representations of as many of the terrain characteristics outlined in ATESv2 (Statham and Campbell, 2023) as possible over capturing very high resolution representations of a few

DEM derivatives.

This holistic perspective of developing an automated ATES model is in line with our approach to reverse engineer the optimal input parameters. Our goal is not to prescribe what an ATES rating should be based on theoretical understanding of avalanche behavior coupled with easily accessible GIS tools. Instead, we are aiming to extract meaningful thresholds from the work of avalanche experts and develop datasets that represent meaningful terrain characteristics for classifying avalanche terrain.

The second likely cause of high resolution DEM input not improving the performance of AutoATES is due to a mismatch in scale between the automated approach and the ATES benchmark maps, as already pointed out by Larsen et al. 2020. Human ATES maps are based on polygon features which are created by manually drawing boundaries around terrain features with similar characteristics (Statham and Campbell, 2023). An advantage of this approach is that it simplifies the ATES map into a set of features that are relevant for the scale of backcountry travel in avalanche terrain. The resulting map provides a smooth

and clean appearance and focuses on the most important features in the terrain. A disadvantage of this approach is that the drawing of polygon boundaries is highly subjective and relies heavily on the local knowledge and experience of the human mapper.

Human maps are also able to consider common travel routes and incorporate overall terrain exposure to capture a bigger picture of recreational use for a given area. For example, mapping an isolated area of simple (1) terrain within an otherwise steep and

highly exposed slope is not necessarily meaningful in the big picture because the routes to access that terrain involve exposing yourself to the surrounding slopes. Incorporating the bigger picture of terrain exposure allows human mappers to consider multiple scales, ranging from terrain features to entire basins, when determining ATES ratings (Statham et al., 2018). The current version of AutoATES is not able to incorporate the cumulative exposure of traveling through avalanche terrain and therefore is purely based on the local terrain characteristics.

Currently, AutoATES attempts to bridge the gap in scale between human and automated maps by including the island filter function to remove small regions in the AutoATES map. This method is effective at simplifying the output of AutoATES and improving alignment with manual maps but results in a loss of data resolution. A fundamental question in developing automated ATES is whether the aim of matching human maps is really the best way to leverage the advantages of a computer based system. While, AutoATES has the capability to produce much higher resolution output compared to manual ATES maps

due to its ability to calculate terrain characteristics on a pixel-by-pixel basis across large study areas, it would be tedious and





impractical for a human ATES mapper to adopt the same approach. This computational advantage could be especially important for zonal ATES mapping where the cumulative hazard across a route is less important than grouping regions with similar characteristics within a larger basin.

## 4.3 Limitations

This research aims to evaluate the performance of the ATES classification portion of the AutoATES model and does not attempt to validate the PRA model and Flow-Py runout model in a robust way. Future research using local avalanche cadastre information could test how changes in the input parameters of the PRA model and runout model impact the overall accuracy of AutoATES. Of particular interest is how the PRA model and runout model perform in maritime snow climates which have so far not been widely tested (D'amboise et al., 2022, 2020).

The study areas selected for this research are both regions with relatively high elevation peaks, steep terrain, and relatively small areas of forested terrain. This terrain is largely representative of the type of mountainous terrain that backcountry skiers recreate in, but it is not necessarily representative of terrain that is favored by other winter recreationists like snowmobilers, snowshoers, hikers, or cross country skiers. Further validation is necessary in areas that are largely non-avalanche (0) terrain or simple (1) terrain with occasional avalanche paths.

## 580 4.4 Practical applications

AutoATES was developed as a tool for large scale automated terrain mapping which can provide avalanche terrain information for much larger swaths of mountainous terrain than previously possible. Using AutoATES effectively requires knowledge of the local snow climate and careful consideration of available DEM and forest input data. AutoATES maps can be used either as a first draft to be manually revised by local experts or as a stand alone product after quality control with local avalanche

experts. Despite the recommendation of vetting the output of AutoATES maps with local experts before release to the public, the automated system still provides massive improvements in efficiency compared to traditional manual mapping methods.

Localizing and validating AutoATES is dependent on the input datasets available. This research uses forest data that likely differs from data available in other provinces or countries. Additional validation and testing of forest input parameters for both the PRA model and ATES classification input parameters is necessary if applying AutoATES in regions with different types

of forest data available.

For those interested in applying AutoATES in new study areas without validation datasets available to localize the model, we recommend starting with the default input parameters presented in Toft et al., 2023. This research has shown that the most likely areas for errors in AutoATES are around treeline elevations where forest datasets may not accurately capture the extent and character of forested terrain. From an end user perspective these are critical areas to capture accurately because of the

frequent use of forested terrain under low visibility conditions or during periods with elevated avalanche hazard. We recommend biasing AutoATES toward overpredicting the ATES rating in these cases so that the risk management practices of end users are biased towards being more conservative. The research presented in this paper has shown that the TREE01 and



OE23 parameters are critical for the accuracy of challenging (2) terrain. We recommend testing a range of input parameters and consulting local avalanche experts to help identify appropriate thresholds to define simple (1) and challenging (2) terrain.

When applying AutoATES in terrain with isolated or small avalanche paths, especially in largely forested terrain, the island filter size is an important parameter to control which features are filtered out. For our study areas the median polygon area from the ATES benchmark maps are 114.2 hectares for non-avalanche (0), 2.7 hectares for simple (1), 49.0 hectares for challenging (2), 407.9 hectares for complex (3), and 16.0 for extreme (4). In comparison the optimized AutoATES island filter size is 30,000 m$^2$ or 3.0 hectares for both Connaught Creek and Bow Summit using the ALOS 26 m DEM. Based on the substantially smaller median size of polygons in simple (1) terrain, it could be advantageous to decrease the island filter size if applying AutoATES in largely forested terrain.

The fact that low resolution DEM data produce AutoATES maps with equivalent accuracy is an advantage for generating ATES maps on large spatial scales. The most computationally expensive component of the AutoATES model chain is the Flow-Py runout model. The computation resources required to process Flow-Py increases as a function of the surface area of the raster cell size, so a 5 m DEM (25 m$^2$) would take roughly four times longer to process compared to a 10 m DEM (100 m$^2$) for the same study area. We recommend using the ALOS 26 m global DEM in areas without higher resolution data available.

## 5 Conclusions

This research aimed to validate and localize a model for generating automated ATES ratings against professional ATES maps in Western Canada and evaluate the impact of different DEM input data. The validation and localization utilized a grid search method to determine the optimal input parameters for two study areas; Bow Summit in Banff National Park, Alberta and Connaught Creek in Glacier National Park, British Columbia. The results demonstrate that the accuracy of AutoATES ranges from 75.1 % to 84.4 % for the two study areas, respectively, as compared to ATES benchmark maps created by three local avalanche experts. In comparison to the three individual manual ATES maps, AutoATES was within 1 percentage point of the average overall accuracy in both study areas. DEM resolution had surprisingly little effect on the output of AutoATES based on four input DEMs in the Connaught Creek study area ranging from 5 m to 26 m resolution. The overall accuracy compared to the ATES benchmark map is 74.5 %, ± 1 % for all DEMs.

Further testing in different snow climates and ecological regions is required to fully understand the best practices for applying AutoATES. Future research should aim to further validate the AutoATES model by specifically validating the PRA model and runout simulation tool Flow-Py. Significant improvements in accuracy and consistency in forested areas are possible with higher resolution forest input data, such as a satellite remote sensing landcover classification or LiDAR (Bebi et al., 2021; Bühler et al., 2022; Sykes et al., 2022). Including a metric for terrain exposure which captures the consequences of falling (Harvey et al., 2018) could complement the avalanche runout exposure metrics currently used in AutoATES, especially for improving accuracy in extreme (4) terrain. Finally, further development of the AutoATES classification model to consider

alternative classification methods such as fuzzy membership models (Veitinger et al., 2016) or machine learning models could
improve accuracy and make future validation efforts more nuanced at highlighting areas where the model has high uncertainty.
A central question in validating automated ATES methods is whether manually generated ATES maps truly represent the gold
standard that automated maps should aim for. The ability of AutoATES to classify terrain on a pixel by pixel basis across large
swaths of mountainous terrain based on a meaningful set of terrain characteristics far exceeds the computational ability of
human mappers. Automated maps have the potential to outperform human maps in terms of precision, replicability, and the
scale that the output can be presented.

An alternative to purely automated ATES mapping, is to leverage the computational efficiency and precision of AutoATES to
generate a first draft map which is then updated based on interpretation of local experts. Combining the strengths of automated
and manual mapping methods provide the optimal trade off between efficiency and capturing local expertise. Producing ATES
maps in this manner could also provide further validation datasets to continue improving AutoATES by characterizing the
differences between the purely automated maps and the manually adjusted versions. To determine the best practices for ATES
mapping in the future, we should consider how to improve end users' ability to make informed risk management decisions
based on the terrain characteristics and current avalanche hazard conditions. Further research is required to test how end users
interpret ATES maps and what characteristics help them make appropriate assessments of avalanche terrain.

AutoATES has demonstrated the ability to classify terrain with similar accuracy to human experts, which presents an
opportunity to redefine how ATES ratings are generated and applied. Currently, maps are only available for high use areas
where resources exist to offset the time and cost of manual mapping. AutoATES opens up the opportunity to create low cost
avalanche terrain information across large areas of mountainous terrain.

**Code and data availability**

The data to replicate the methods in this manuscript are available in our Open Science Framework repository (Sykes et al.,
2023). AutoATES is open source and the model code is available via GitHub (https://github.com/AutoATES/AutoATES-
v2.0).

**Author contributions**

HT was the creator of AutoATES and continues to maintain and develop it. JS contributes to maintenance and development
of AutoATES and carried out the validation and localization analysis with support from PH. GS developed the ATES and
ATESv2 models and helped create the validation dataset for this research with support from Avalanche Canada and Parks
Canada. JS prepared the manuscript with PH as the primary editor. All co-authors contributed to the final manuscript.



**Competing interests**

PH is a member of the editorial board of Natural Hazards and Earth System Sciences. The authors do not declare any other competing interests.

**Acknowledgements**

The study areas of this research are located on the ancestral and unceded territories of Canadian First Nations; including the Blackfoot, Ktunaxa, Metis, Okanagan, Secwépemc, Sinixt, Stoney, and Tsuut'ina people. We wish to acknowledge our collaborators on this research from Avalanche Canada and Parks Canada, specifically Karl Klassen for the initial motivation for the research. Thanks to Andrew Schauer from Chugach National Forest Avalanche Center for his contributions towards

the development of AutoATES and feedback on this manuscript. The NSERC Industrial Research Chair in Avalanche Risk Management at Simon Fraser University is financially supported by Canadian Pacific Railway, HeliCat Canada, Mike Wiegele Helicopter Skiing and the Canadian Avalanche Association. The research program receives additional support from Avalanche Canada and the Avalanche Canada Foundation.

**Financial support**

This research was funded by the Government of Canada Natural Sciences and Engineering Research Council via the NSERC Industrial Research Chair in Avalanche Risk Management at Simon Fraser University (grant no. IRC/515532-2016).





**Figure A1: Grid Search Plot for the Canadian 17 m DEM in Connaught Creek.**



**Figure A2: Grid search plots for Stereo 10 m DSM in Connaught Creek.**






**Figure A3: Grid search plots for LiDAR 5 m DTM in Connaught Creek.**





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
