# Peer review of "Automated Avalanche Terrain Exposure Scale (ATES) mapping -Local validation and optimization in Western Canada"

_Natural Hazards and Earth System Sciences, 2023_

## Referee Comment (RC2)

**Line comments (non-exhaustive)**

- Table 1: Why is some content in this table in bold font? 'events' appears to be underlined in blue (?)
- Figure 1: The location of the site in the inset is a bit hard to make out – maybe drop the black outline and chose a brighter fill colour for the polygon (and/or chose a smaller scale)
- Line 135: well-defined
- Figure 3: In upper left, move 'b) Forest' to not be covered by dotted line; in lower left, b) ATES Map 3 → c) ATES Map 3
- Line 195: [...] ATESv2 (Table 1).
- Line 260ff: To keep things clear, I advise sticking to the specific terms (DTM and DSM) introduced in lines 120f, rather than the generic 'DEM', in lines 260ff and below (e.g. in Figure 6 figure caption)
- Line 287: [...] row of Figures 7 and 8
- Line 289: [...] terrain (Figures 7 & 8)
- Line 292: [...] causes

---

## Author Comment (AC1)

**Response to reviewer comments**

Sykes, J., Toft, H., Haegeli, P., and Statham, G.: Automated Avalanche Terrain Exposure Scale (ATES) mapping – Local validation and optimization in Western Canada, Nat. Hazards Earth Syst. Sci. Discuss. [preprint], https://doi.org/10.5194/nhess-2023-112, in review, 2023.

**Reviewer 1 – Zachary Miller**

**Overall**

Accepted subject to minor revisions.

The manuscript titled "Automated Avalanche Terrain Exposure Scale (ATES) Local validation and optimization in Western Canada" explores a relevant comparison of automated and human derived avalanche hazard mapping. It leverages the updated AutoATESv2 processing workflow with specific reverse-engineering techniques to produce maps of two Canadian domains and quantifies the accuracy of those maps against human created maps that leverage expert knowledge of specific terrain contained within the two domains. The work also quantifies the effects of DEM resolution on ATES classification accuracy for the first time. This research is a valuable contribution to the snow avalanche natural hazards research, forecasting, and communications communities.

**General Comments**

**1.1 ATES Color Palette**

The primary concern I have with the manuscript is the color palette chosen for mapping not being red-green (Protanopia) colorblind friendly. I realize this color choice is previously established in ATES mapping, but differentiating between the "Simple" and "Extreme" polygon colors (and Figure 11 bars) is not easy and could lead to considerable confusion in the interpretation of this very valuable tool and the manuscript. I appreciate that the authors utilized differentiated line types in their grid search results figures (7 & 8 and appendix). Please consider adjusting this before final publication.

**Author response:**
**The design of an updated color palette for ATESv2 is an open research question that has been the focus of some prior research, but a suitable solution is yet to be finalized. We appreciate this comment and desire to make our research as accessible as possible to readers with color blindness. While a complete redesign of the ATESv2 color palette is beyond the scope of this manuscript, we have updated the existing color palette to use more color-blind friendly versions of the existing (white, green, blue, black, red) color**

**palette. After testing both the initial version and the updated colors using the Coblis Color Blindness Simulator, we believe the new color scale improves accessibility significantly. We plan to update all figures and tables to implement the new color scale.**

**Original color palette on first row and updated color palette on second row**

| #dadada | #1b6436 | #21298f | #343434 | #ce0014 |
|---|---|---|---|---|
| #ffffff | #28c900 | #007bff | #000000 | #ff0138 |

**Specific comments**

- Line 102 – "Figures 2 & 3" should be "Figures 1 & 2."
    - **Author Response:** We will correct this typo in the revised manuscript.
- Line 116 – The authors mention Rogers Pass without any clarification as to where it is in relation to study site (Connaught Creek) or marking on study site maps. Please clarify for readers without prior knowledge.
    - **Author Response:** We added 'is located at the summit of Rogers Pass' to line 107 to clarify the location relative to the study area.
- Line 119 – What years are the DEM/DSM data from? The fine-resolution surface elevations and vegetation cover may change frequently in such a dynamic environment and be relevant to the outputs of the ATES analysis.
    - **Author Response:** We have reached out to Parks Canada GIS Staff to determine the years that the DEM data were created and plan to include this information in the updated manuscript.
- Line 153-154 – The numbering of basic "processing" doesn't align with the referenced plot (Figure 3). Update to include "Runout simulation" within step 1 as shown in figure and include "Validation data generation" as step 2, or adapt figure?
    - **Author Response:** This is an excellent suggestion, and we will update Figure 3 to better align with the description in section 2.2, lines 154-159.
- Figure 3 – Great plot but imagery and text boxes/lines are blurry. Perhaps re-produce at higher dpi.
    - **Author Response:** We will increase the resolution of figure 3 in the updated manuscript.
- Line 245 – Please briefly define "F1 score," "precision," and "recall" to clarify their relevance as quality metrics.
    - **Author Response:** We elaborated on the description of these performance metrics and our justification for including them in the manuscript on lines 245-247. A more detailed description of these performance metrics seems outside the main focus of the manuscript.

- Line 287 & 289 – "Figures 8 & 9" should be "Figures 7 & 8."
  - **Author Response:** We will correct this typo in the revised manuscript.
- Line 304 – "Figure 7,8" should be "Figures 7 & 8" to keep consistent with style.
  - **Author Response:** We will correct this inconsistency in the revised manuscript.
- Figures 7 & 8 – "Optimized value" symbol in legend is a grey vertical line and is a light blue vertical line in the plots – fix to match.
  - **Author Response:** This mismatch of colors will be fixed in the updated manuscript.
- Line 433 – remove "26m" from description of the ALOS DEM to match other referenced DEM/DSMs in sentence.
  - **Author Response:** Our intention in specifying 26m with the ALOS DEM is to clarify which version of the ALOS DEM products we are using. However, we agree that this information does not need to be specified in each instance that we mention the ALOS DEM. We will update the manuscript to only specify the resolution of the ALOS DEM in the same context as the other DEM data used in the study.

**Reviewer 2 – Marc Adams**

**Overall**

Accepted subject to minor revisions.

In this contribution, an automised procedure is evaluated, which allows classifying mountainous terrain based on the degree to which backcountry recreationists are exposed to avalanche hazard. The Automated Avalanche Terrain Exposure Scale (ATES) goes back to 2006, when first maps were manually drafted by (local) avalanche experts with the aim of providing decision-support to folks traveling in wintry mountainous terrain. Against the backdrop of an increasing demand for ATES-maps from the community and the necessity to be able to map larger areas at lower costs, an automated routine was developed. This routine uses GIS tool chains and open-source avalanche simulation models. In this manuscript, the routine is validated against manually drafted maps and a site-specific sensitivity study of the key parameters for map generation carried out and analysed.

Overall, the manuscript is clearly structured, succinct and very well written - a joy to read! Figures and Tables are well presented and organised, captions are clearly written and comprehensible.

**General Comments**

**2.1 Effect of DEM type and resolution on AutoATES accuracy**

Apart from a few minor typos, consistency issues and glitches in some figures (see line comments below), I was hard-pressed to find any points to critique in the manuscript. One thing that did strike me, was that I would have expected the impact of using different types of DEM (DTM and DSM) and different resolution DEMs to be much higher on the calculated results. Did I understand correctly that for Connaught Creek both DSM (10m, 17m & 26m from different sources) and DTM (LiDAR) were used in the AutoATES routine, i.e. one bare-earth dataset and three datasets physically including trees? I can see how differing grid resolutions would not impact the result much (especially as forest input data remained the same, as pointed out in the manuscript), but the physical presence of the trees in the DSM, I would have thought to impact for example the FlowPy simulations and thus overhead exposure. This point is however more of a general observation on my part rather than a shortcoming of the manuscript.

**Author response: We were also surprised by the lack of improved accuracy with the high resolution DEM data and the lack of differences in ATES output based on DEM type. The most likely explanation for this is that we did not include the PRA or runout model parameters as part of our grid search due to computer processing limitations and lack of available validation data. Future research should aim to optimize the PRA model and runout model (Flow-Py) parameters in the validation process in order to develop a better understanding of DEM sensitivity. We discuss these results and limitations in sections 4.2.1 and 4.3 as well as in the conclusion on *lines 620-622.***

**Specific comments**

- Table 1: Why is some content in this table in bold font? 'events' appears to be underlined in blue (?)
    - **Author Response:** The bold text is part of the system for designating the ATES rating for specific terrain features. To clarify we added 'Criteria highlighted in bold indicate default values that automatically place the ATES rating into that category or higher' to table caption. Line 67-68
- Figure 1: The location of the site in the inset is a bit hard to make out – maybe drop the black outline and chose a brighter fill color for the polygon (and/or chose a smaller scale)
    - **Author Response:** Thank you for bringing this to our attention. We updated the figure by choosing a smaller scale, decreasing the width of the black outline, and choosing a brighter red fill color.
- Line 135: well-defined
    - **Author Response:** We will correct this typo in the revised manuscript.
- Figure 3: In upper left, move 'b) Forest' to not be covered by dotted line; in lower left, b) ATES Map 3 -> c) ATES Map 3
    - **Author Response:** Figure 3 has been updated to address these concerns and those of the other reviewer.

- Line 195: […] ATESv2 (Table 1).
  - **Author Response:** We will correct this typo in the revised manuscript.

- Line 260ff: To keep things clear, I advise sticking to the specific terms (DTM and DSM) introduced in lines 120f, rather than the generic 'DEM', in lines 260ff and below (e.g. in Figure 6 figure caption)
  - **Author Response:** We agree with this suggestion and have updated the terminology in the manuscript to reflect the DEM type (DTM vs DSM).

- Line 287: […] row of Figures 7 and 8
  - **Author Response:** We will correct this typo in the revised manuscript.

- Line 289: […] terrain (Figures 7 & 8)
  - **Author Response:** We will correct this typo in the revised manuscript.

- Line 292: […] causes
  - **Author Response:** We will correct this typo in the revised manuscript.